# Impact of rising temperatures on the bacterial communities of *Aphaenogaster* ants

**Lily A. Kelleher[1,2,*] and Manuela O. Ramalho[1]**

## ABSTRACT

Studies have shown that biodiversity will be impacted by global climate change, with the effect on ants just beginning to be documented. The influence on ant symbiotic bacterial communities remains understudied. *Aphaenogaster* Mayr, 1853, are seed-dispersing ants in deciduous forests and their bacterial communities have just been uncovered; however, much is unknown. We aim to determine the impact that warming temperatures will have on *Aphaenogaster* survival and on their bacterial communities. Ants from four colonies were collected from West Chester, PA, USA and entire colonies were subjected to a control temperature (22°C). After 6-12 months, the same colonies were subjected to an experimental temperature (32°C). DNA was then extracted from ants of all development stages and the 16S rRNA gene was amplified and sequenced following the NGS amplicon approach. The findings revealed that *Aphaenogaster* ant mortality rates increased, and their symbiotic bacterial communities changed in warmer temperatures. This resulted in a decrease in the presence of *Wolbachia* spp. and an increase in the presence of *Corynebacterium* sp. This study reveals important information about the impact of warming temperature on *Aphaenogaster* ants, and we suggest methods to help protect these ants and other insects in the future.

KEY WORDS: Ecology, Host–bacterial interactions, Climate change, Heat stress, *Wolbachia*, 16s rRNA Sequencing

## INTRODUCTION

It is important to understand the influence of temperature on biodiversity because of global climate change. Climate change is causing extreme weather events and changes in temperature across the globe (Huber and Gulledge, 2017; Kundzewicz, 2016; Adedeji et al., 2014). This is impacting global biodiversity by threatening habitats and impacting the behavior and range of various organisms (Omann et al., 2009; Gonzalez-Orozco et al., 2016; Garcia et al., 2014; Suárez et al., 2002; Magozzi and Calosi, 2015). An important contribution to our planet's biodiversity comes from host–microbe interactions; however, few publications have focused on the influence of climate change on these relationships. These studies have found that climate change will cause shifts in the bacterial community compositions of birds (Zhu et al., 2019), mammals (Tajima et al., 2007), reptiles/amphibians (Fontaine et al., 2018; Kohl and Yahn, 2016), and aquatic animals (Diwan et al., 2023; Scanes et al., 2021). Unfortunately, many organisms with essential host–bacterial interactions have been left out of these studies, such as ants.

Ants are ectotherms, which makes them a thermally responsive taxa and in certain ants, such as *Acromyrmex heyeri*, their ecological performance increases with temperature to a certain point before rapidly declining (Scudder, 2017; Bollazzi et al., 2008). Ants can regulate the microclimate within their nests, which has led to them being left out of many climate change studies because it was thought that they would not be significantly impacted by changes in temperatures. However, a study by Youngsteadt et al. (2023) showed that there is a limit to ants' ability to thermoregulate the microclimate within their nests and they will ultimately be influenced by climate change (Youngsteadt et al., 2023). Additionally, some studies that have looked at the influence of climate change on ants have shown that it will impact the metabolism, development and performance of various ants, including *Crematogaster lineolata*, *Aphaenogaster* spp., *Camponotus* spp., *Myrmica* spp., and *Temnothorax longispinosus* (Parr and Bishop, 2022; Diamond et al., 2016).

Many ant genera (ie. *Camponotus*, *Cephalotes*, *Daceton*, etc.) rely strongly upon their symbiotic relationships, especially with plants, fungi and bacteria (Engel and Moran, 2013; Moreau, 2020; Hu et al., 2018; Lester et al., 2017; McMunn et al., 2022; Ramalho et al., 2017a, 2020). Some species of ants rely on the diversity and functionality of their symbiotic communities to perform basic biological functions, such as nutrient acquisition of nitrogen and amino acids, defense against pathogens and environmental tolerance (Engel and Moran, 2013; Moreau, 2020; Hu et al., 2018). Hence, any depletion or disruption to the symbiotic ant bacterial communities can be detrimental to both ants and the ecosystems upon which they reside. This disruption can affect various ecological processes, such as soil aeration, nutrient cycling, seed dispersal, pest control, decomposition, and the overall balance of the ecosystem (Lester et al., 2017). Some factors that influence ant bacterial communities are genetics, diet, social castes, temperature, and geography (Diamond et al., 2016; Engel and Moran, 2013; McMunn et al., 2022; Ramalho et al., 2017a, 2020).

Given the significance of the symbiotic bacterial relationships that ants, such as *Camponotus*, *Cephalotes*, *Formica*, and *Solenopsis* engage in, it is important to understand the impact of changing temperatures on the ant bacterial communities. Bacterial community composition in various environments, such as oceans, hot springs and soils, and within ants, such as *Cephalotes*, have been shown to be impacted by temperature variations due to each bacteria having varying temperature sensitivities (McMunn et al., 2022; Ramalho et al., 2020; Dal Bello and Abreu, 2024; Donhauser et al., 2020; Uribe-Lorío et al., 2019). This impact on bacteria can in turn impact their ant hosts due to the reliance of some ants on their symbiotic communities. Previous studies have shown that *Cephalotes* microbial communities are different based on temperature regions and the

[1]Department of Biology, West Chester University, West Chester, PA, USA, 19383. [2]Department of Evolution, Ecology, and Organismal Biology, The Ohio State University, Columbus, OH, USA, 43210.

*Author for correspondence (kelleher.144@osu.edu)

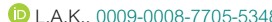 L.A.K., 0009-0008-7705-5348

adaptation to environmental changes and behavioral plasticity of ants will not protect them from the warming climate (McMunn et al., 2022; Penick et al., 2017). As the habitats that ants reside in warm, metabolic changes due to shifts in microbial communities will be experienced by various species, such as: *Solenopsis invicta*, *Brachyponera chinensis*, *Camponotus castaneus*, *Camponotus chromaiodes*, *Camponotus pennsylvanicus*, and *Formica subsericea* (Youngsteadt et al., 2023; Xiao et al., 2023).

This study will focus on better understanding how the bacterial community composition and abundance of *Aphaenogaster rudis* is impacted by changing temperatures, being the first study of its type to focus on this ecologically important ant genus. *Aphaenogaster* is a globally distributed genus and still remains largely understudied in terms of their bacterial community. Previous studies have shown that changing temperatures can impact the range, food consumption, respiration, enzymatic activity and heat shock response of *Aphaenogaster* species (Penick et al., 2017; Lau et al., 2019; Miller, 2018; Warren and Chick, 2013; Cahan et al., 2017; Hofmann and Todgham, 2010). However, the *Aphaenogaster* bacterial community is largely undocumented, and little is known about the impact that global climate change will have on their bacterial communities (Pagalilauan et al., 2025; Kelleher and Ramalho, 2025). If climate change causes a modification of the bacterial community composition of *Aphaenogaster* and causes changes in behavior and mortality rates, this can help reveal how important some of these microbes are to them and enhance our understanding of the host–microbe interactions in response to different temperature conditions.

By studying the impact of global climate change on the host–bacterial interactions of ants we can try to predict how these interactions will evolve in the coming years. Using this information, we can implement methods for remediation and conservation of areas where these threatened ants and other organisms are located in an attempt to mediate the impact of climate change and protect our biodiversity. The questions this study aims to answer are: How do warming temperatures impact the survival of *Aphaenogaster* ants? How do warming temperatures impact the bacterial community of *Aphaenogaster* ants? We hypothesize that *Aphaenogaster* mortality rates will begin to increase as temperatures rise and that there will be a shift in their bacterial community composition. By addressing these questions, we can gain a comprehensive understanding of how global climate change will impact these important seed dispersing ants and we can predict how host–bacterial interactions will change in the future. These findings can inform the development of restoration and conservation strategies aimed at minimizing the ecological impact of these environmental changes on ants and other insect communities.

## RESULTS
### Behavioral observations
Ants were stored in a temperature-controlled incubator over the course of 98 days. The temperature was slowly raised from 23°C to 32°C. From 23°C to 31°C the mortality rates within the colonies were consistent, averaging at 1.38±1.46 ants per day (Table S2). This mortality rate is similar to that of colonies where temperature was maintained at 22°C, which averaged 1.69±1.05 ants per day. However, once 32°C was reached, there was a spike in mortality for all colonies with an average mortality of 8.50±10.41 ants per day.

A logistic regression model showed that the rate of mortality increase over time was significantly greater in the four colonies subjected to warming compared to the one colony raised at 22°C. We acknowledge that only one *Aphaenogaster* colony was used as a control for this analysis, which may limit the robustness of the comparison and skew the results. Specifically, Colony A had a significantly faster mortality rate than the Control (Logistic regression model: *P*-value=0.00285), Colony B had a significantly faster mortality rate than the Control (*P-value*=0.000341), Colony C had a significantly faster mortality rate than the Control (*P*-value=0.00175), and Colony D had a significantly faster mortality rate than the Control (*P*-value<$2\times10^{-16}$). At day 71 there was observed a spike in mortality rates of the four experimental colonies (Fig. 1: red dotted line).

### Alpha diversity analysis
The bacterial community of the *A. rudis* samples contained 7951 bacterial ASVs (Fig. 2; Fig. S2). A core bacterial community

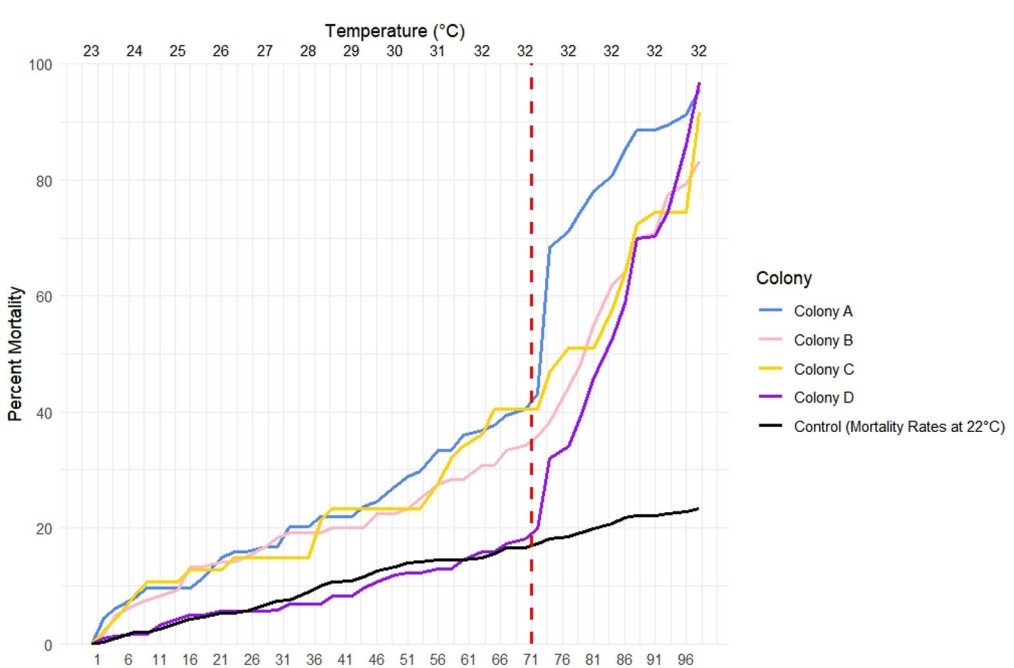

**Fig. 1. Percent mortality of workers over the course of the experiment.** X-axis displays day and Y-axis displays percent mortality. Each line represents a different colony or the control. The vertical red line indicated the point where there was a spike in mortality rates.

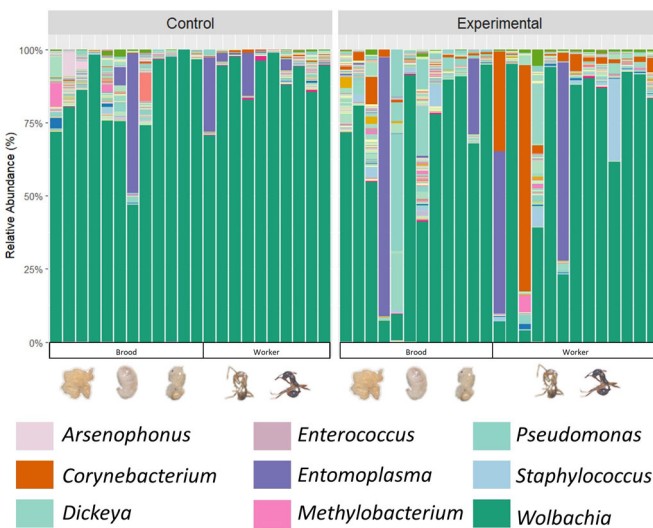

**Fig. 2. Stacked bar cart of the whole bacterial community (ASVs) at the genus level found in *Aphaenogaster rudis*.** Each column represents a sample, and each color represents a bacteria genus (ASV). The legend depicts the ten most abundant genera within the bacterial community. Other genera can be observed in Fig. S2. There was found to be a total of 7951 bacterial ASVs in the bacterial communities of the *Aphaenogaster* samples.

analysis was run to identify any bacteria that were occurring in more than 50% of the samples. The core bacterial community analysis for all samples revealed that the *A. rudis* in this study do have a core bacterial community. The overall core bacterial community that was identified in the *A. rudis* samples consisted of 29 ASVS which comprised 2 genera (Fig. 3A; Percent relative abundance in core community): *Wolbachia* (97.79%) and *Corynebacterium* (2.21%).

A core bacterial community was also identified for both the control and experimental groups separately to account for possible differences in their bacterial communities. A core bacterial community was identified in the *A. rudis* samples for the control that consisted of 29 ASVS which comprised three genera (Fig. 3B; percent relative abundance in core community): *Wolbachia* (94.19%), *Entomoplasma* (4.41%), and *Weissella* (1.40%). A core bacterial community was identified in the *A. rudis* samples from the experimental group that consisted of 48 ASVs that comprised seven genera (Fig. 3C; percent relative abundance in core community): *Wolbachia* (90.26%), *Corynebacterium* (5.70%), *Staphylococcus* (1.83%), *Hydrocarboniphaga* (0.83%), *Methylobacterium-methlonbrum* (0.62%), *Acinetobacter* (0.58%), and *Sphingobium* (0.27%).

Within the whole bacterial community, there was not a significant difference in species richness and evenness between the treatments (Shannon diversity: $H=4.025$, $P$-value$=0.403$). However, species richness and evenness were high and the bacterial communities of all the samples exhibited high species diversity with low dominance (Margalef's index: $D=21.82\pm13.58$; Simpson's dominance index: $D=0.047$).

## Influence of temperature treatment on the bacterial communities

Beta diversity analysis, using unifrac distance matrices, revealed that the composition of the bacterial communities of the control and the experimental group are significantly different [Permutational multivariate analysis of variance (PERMANOVA) unweighted unifrac distance: $P$-value$=0.008$; $pseudo$-$F=2.187$;

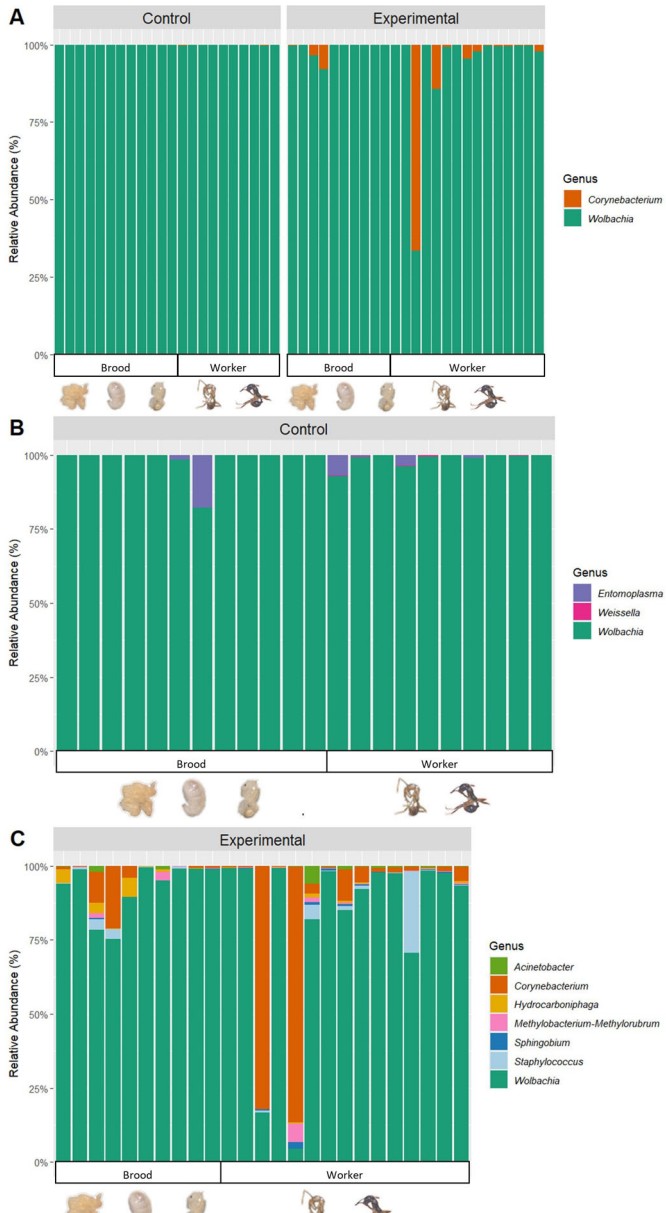

**Fig. 3. Stacked bar charts of the core bacterial communities (ASVs) of *Aphaenogaster rudis*.** (A) Stacked bar chart of the core bacterial communities (ASVs) at the genus level found in *A. rudis* with 16 s rRNA amplicon sequencing. There were three bacteria genera in the core bacterial community of all *Aphaenogaster* samples. (B) Stacked bar chart of the core bacterial communities (ASVs) at the genus level found in the *A. rudis* control with 16 s rRNA amplicon sequencing. There were three bacteria genera in the core bacterial community of the *Aphaenogaster* control samples. (C) Stacked bar chart of the core bacterial communities (ASVs) at the genus level found in *A. rudis* experimental group with 16s rRNA amplicon sequencing. There were eight bacteria genera in the core bacterial community of the *Aphaenogaster* experimental group samples. Each column represents a sample, and each color represents a bacteria genus (ASV).

Fig. 4; Figs S3 and S4]. This is depicted in the non-metric multidimensional scaling (nMDS) diagram (Fig. 4).

SIMPER analysis was performed to look at the main bacteria (ASVs) associated with the control and experimental groups (Table 1 and Fig. 5). *Wolbachia* spp. were shown to be driving the significant difference between the control and experimental treatments. The top

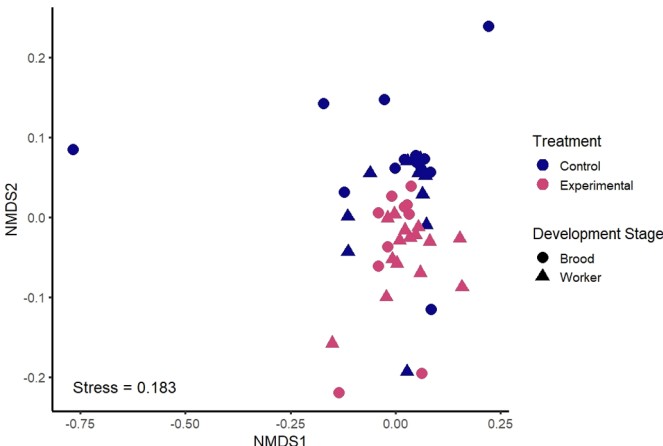

**Fig. 4. Non-metric multidimensional scaling diagram (nMDS) using Bray–Curtis dissimilarity matrix.** The closer the dots are to one another the more similar the composition and abundance of the bacterial communities. Dots are colored by treatment and shaped by development stage.

bacteria that contributed to this significant difference were (% contribution to overall difference): *Wolbachia* sp. 1 (4.95%), *Wolbachia* sp. 2 (4.16%), *Wolbachia* sp. 3 (3.97%), *Wolbachia* sp. 4 (3.40%), *Wolbachia* sp. 5 (3.20%), *Wolbachia* sp. 6 (3.09%), and *Wolbachia* sp. 7 (2.93%). All seven of these bacteria were in higher abundance in the control than the experimental treatment.

### Influence of colony and development stage on the bacterial communities

There was not a significant difference between the bacterial community composition and abundance of the different colonies (Colony A-D) (PERMANOVA unweighted unifrac distance: *P*-value=0.217; *pseudo-F*=1.148; PERMANOVA weighted unifrac distance: *P*-value=0.321; *pseudo-F*=1.113). Additionally, this analysis was performed for both the control and experimental treatments separately to account for any influence that treatment had (control: PERMANOVA unweighted unifrac distance: *P*-value=0.112; *pseudo-F*=1.264; PERMANOVA weighted unifrac distance: *P*-value=0.273; *pseudo-F*=1.192; experimental: PERMANOVA unweighted unifrac distance: *P*-value= 0.46; *pseudo-F*=1.00; PERMANOVA weighted unifrac distance: *P*-value=0.519; *pseudo-F*=0.868).

There was a significant difference in the composition and abundance of the bacterial communities based on development stage (PERMANOVA weighted unifrac distance: *P*-value=0.032; *pseudo-F*=2.123). Post-hoc pairwise PERMANOVA tests (weighted unifrac distance) were performed to analyze each developmental stage in pairs (Table S3). No pairs showed significance, possibly due to the low significant power due to the small number of samples when

looked at in this manner. Instead, we analyzed brood (eggs, larvae, and pupa) versus workers (recently emerged workers and workers). There was a significant difference in the bactureerial community composition and abundance of the brood and workers (PERMANOVA weighted unifrac distance: *P*-value=0.03; *pseudo-F*=3.368; Fig. 4; Figs S3 and S4).

SIMPER analysis was performed on the bacterial community to look at the main bacteria (ASVs) associated with the significant result from the brood versus workers PERMANOVA test (Table 2). *Wolbachia* spp. were shown to be driving the significant difference between the workers and brood. The following bacteria were in higher abundance in workers than brood (% contribution to overall difference): *Wolbachia* sp. 1 (4.91%), *Wolbachia* sp. 3 (4.09%), *Wolbachia* sp. 4 (3.37%), *Wolbachia* sp. 5 (3.18%), and *Wolbachia* sp. 6 (3.06%). The following bacteria were in higher abundance in brood than workers (% contribution to overall difference): *Wolbachia* sp. 2 (4.17%) and *Wolbachia* sp. 7 (2.94%).

Samples were separated by treatment to see if the brood and workers were impacted differently at each treatment. For the control there was not a significant difference in the bacterial community composition and abundance for the brood and workers (PERMANOVA unweighted unifrac distance: *P-value*=0.459; *pseudo-F*=0.957; PERMANOVA weighted unifrac distance: *P*-value=0.093; *pseudo-F*=1.959). For the experimental group there was a significant difference in the composition of the bacterial communities of the brood and workers (PERMANOVA unweighted unifrac distance: *P*-value=0.044; *pseudo-F*=1.541). Additional analyses were run to see if brood from the control and experimental treatments were different and if workers from the control and experimental treatments were different. There was not a significant difference between the bacterial community composition and abundance of the brood from the control and experimental treatments (PERMANOVA unweighted unifrac distance: *P*-value=0.134; *pseudo-F*=1.334; PERMANOVA weighted unifrac distance: *P*-value=0.197; *pseudo-F*=1.384). There was not a significant difference between in the bacterial community composition and abundance of the workers from the control and experimental treatments (PERMANOVA unweighted unifrac distance: *P*-value=0.088; *pseudo-F*=1.538; PERMANOVA weighted unifrac distance: *P*-value=0.23; *pseudo-F*=1.479).

SIMPER analysis was performed on the bacterial community to look at the main bacteria (ASVs) associated with the significant result from the brood versus worker PERMANOVA test from the experimental group (Table 3). *Wolbachia* spp. were shown to be driving the significant difference between the workers and brood. The following bacteria were in higher abundance in workers than brood (% contribution to overall difference): *Wolbachia* sp. 1 (3.94%), *Wolbachia* sp. 2 (3.19%), and *Wolbachia* sp. 4 (2.68%). The following bacteria were in higher abundance in brood than

**Table 1. Similarity percentage (SIMPER) analysis of main bacteria in each treatment**

| Comparison | Overall average dissimilarity | Most influential ASV/taxonomy | Percent contribution to difference (%) | Cumulative percent (%) |
|---|---|---|---|---|
| Treatment (control versus experimental) (*P-value*=0.001; *pseudo-F*=2.543) | 68.44% | *Wolbachia* sp. 1 | 4.96 | 4.96 |
| | | *Wolbachia* sp. 2 | 4.16 | 9.12 |
| | | *Wolbachia* sp. 3 | 3.97 | 13.09 |
| | | *Wolbachia* sp. 4 | 3.40 | 16.50 |
| | | *Wolbachia* sp. 5 | 3.20 | 19.70 |
| | | *Wolbachia* sp. 6 | 3.09 | 22.79 |
| | | *Wolbachia* sp. 7 | 2.93 | 25.72 |

Analysis allowed for determination of which ASVs were contributing to the difference in community composition among treatments. *Wolbachia* spp. contributed the most to differences in community composition.

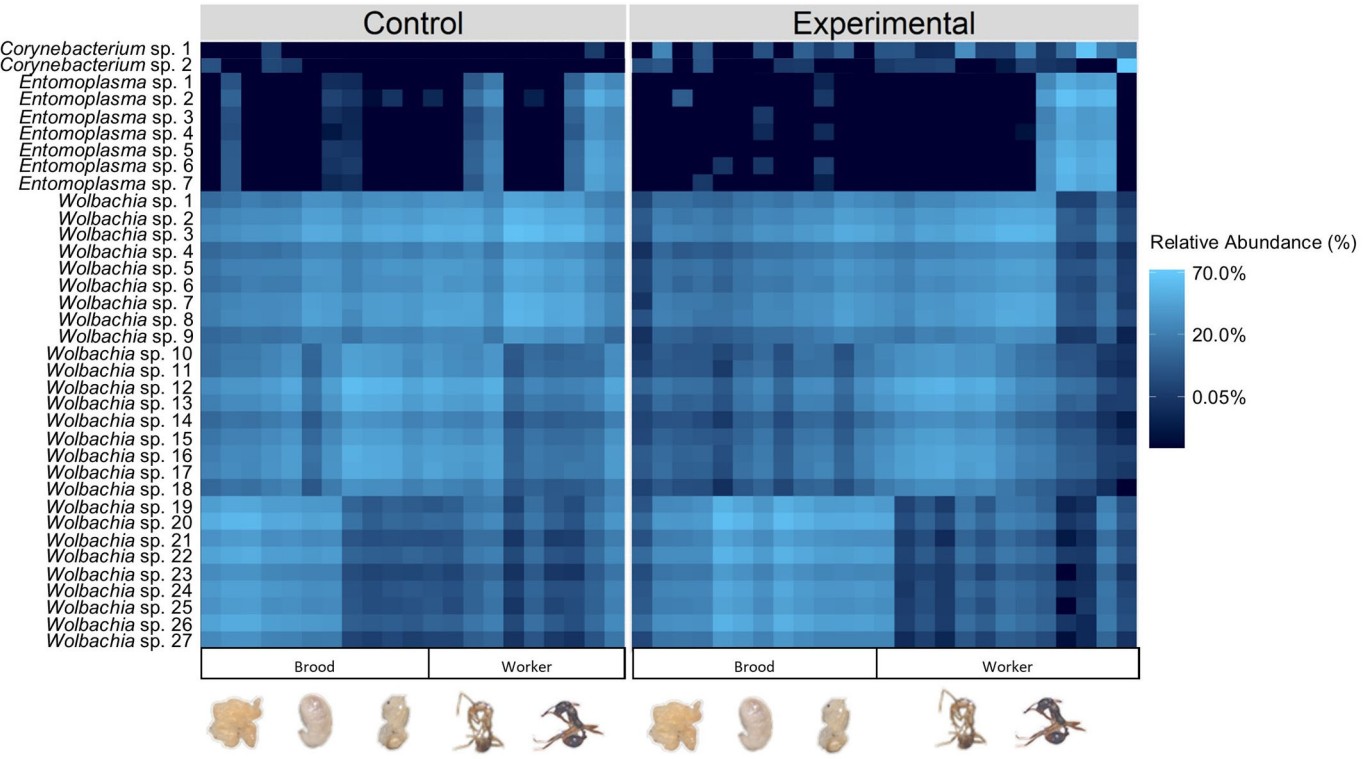

**Fig. 5. Relative abundance of bacteria associated with each treatment and brood or workers.** This is a subset of 36 ASVs that had an average abundance greater than 0.05% across all samples. The shading indicates variation in the relative abundance of different bacteria in all samples ranging from low abundance (navy/black) to high abundance (light blue).

workers (% contribution to overall difference): *Wolbachia* sp. 3 (4.51%), *Wolbachia* sp. 8 (3.12%), *Wolbachia* sp. 9 (2.99%), and *Wolbachia* sp. 10 (2.83%).

## DISCUSSION

This study sought to predict the impact that warming temperatures due to global climate change will have on ant survival and their symbiotic bacterial communities. This is the first study to investigate the impact of warming temperatures on the bacterial communities of *A. rudis* – an important native from the Nearctic region and seed dispersing ant (Penick et al., 2017; Lau et al., 2019; Miller, 2018; Warren and Chick, 2013; Cahan et al., 2017; Hofmann and Todgham, 2010; Pagalilauan et al., 2025; Kelleher and Ramalho, 2025). The results of this study determined that mortality rates within *A. rudis* colonies increased as temperatures began to rise and we observed less brood present after 32°C was reached. Additionally, the bacterial communities of *A. rudis* before and after being subjected to warming temperatures were different with a higher presence of *Wolbachia* at lower temperatures and a higher presence of *Corynebacterium* at higher temperatures.

### Impact of warming temperatures on *A. rudis* survival

The results of this study show that *A. rudis* mortality rates increase 4-5× after 32°C was reached compared to mortality rates at 22°C. A previous study on *A. rudis* looked at the impact of the temperatures, 20°C, 23°C, 26°C, and 29°C on *A. rudis* colony mortality. They found that the mortality rate of the colonies of *A. rudis* increased as temperatures increased. More specifically they found a three times higher mortality rate for colonies at 29°C compared to the colonies at other temperatures, congruent with the rest of our study (Penick et al., 2017). The increased mortality rate in *A. rudis* colonies could be correlated with some physiological changes. Studies have found that warmer temperatures cause early onset senescence in *Aphaenogaster* (Southerland, 1988) and the heat shock response in *Aphaenogaster* is not well adapted to long-term climate variations (Cahan et al., 2017). However, there is potential that

**Table 2. Similarity percentage (SIMPER) analysis of main bacteria in brood or workers**

| Comparison | Overall average dissimilarity | Most influential ASV/taxonomy | Percent contribution to difference (%) | Cumulative percent (%) |
|---|---|---|---|---|
| Brood versus workers for all samples (*P-value*=0.030; *pseudo-F*=3.368) | 69.24 | *Wolbachia* sp. 1 | 4.91 | 4.91 |
| | | *Wolbachia* sp. 2 | 4.17 | 9.08 |
| | | *Wolbachia* sp. 3 | 4.09 | 13.17 |
| | | *Wolbachia* sp. 4 | 3.37 | 16.54 |
| | | *Wolbachia* sp. 5 | 3.18 | 19.72 |
| | | *Wolbachia* sp. 6 | 3.06 | 22.77 |
| | | *Wolbachia* sp. 7 | 2.94 | 25.71 |

Analysis allowed for determination of which ASVs were contributing to the difference in community composition between developmental stages of all samples. *Wolbachia* spp. were determined to contribute the most to differences in community composition.

**Table 3. Similarity percentage (SIMPER) analysis of main bacteria in brood or workers within the experimental group**

| Comparison | Overall average dissimilarity | Most influential ASV/taxonomy | Percent contribution to difference (%) | Cumulative percent (%) |
|---|---|---|---|---|
| | | *Wolbachia* sp. 3 | 4.51 | 4.51 |
| | | *Wolbachia* sp. 1 | 3.94 | 8.45 |
| | | *Wolbachia* sp. 2 | 3.19 | 11.64 |
| Brood versus workers for experimental treatment samples (*P-value*=0.044; *pseudo-F*=1.541) | 73.08 | *Wolbachia* sp. 8 | 3.12 | 14.75 |
| | | *Wolbachia* sp. 9 | 2.99 | 17.74 |
| | | *Wolbachia* sp. 10 | 2.83 | 20.58 |
| | | *Wolbachia* sp. 4 | 2.68 | 23.26 |

Analysis allowed for determination of which ASVs were contributing to the difference in community composition between developmental stages of samples from the experimental treatment. *Wolbachia* spp. were determined to contribute the most to differences in community composition.

changes in their bacterial community composition could be associated with their mortality rates.

The behavior of individual colonies also remained consistent throughout the experiment. Ants were highly active and foraging. After 30°C was reached we observed that Colony C began frequently moving their brood to new locations and brood were observed in a new location every other day. This can be interpreted as a 'stressed' response. No other colonies showed changes in behavior. After temperatures reached 32°C the brood within all colonies drastically decreased and pupa were no longer present in any colonies. However, this data was based on daily observations made on the colonies and the scope of our study was not to investigate the influence of warming temperatures on brood and brood care. This information is inconclusive but raises important questions for future studies about the influence of warmer temperatures on brood survival.

Additionally, we observed (no compositional data was collected) an increase in fat storage in the larvae of *A. rudis* in the experimental treatment compared to the control treatment (Fig. S1). Previous studies have shown many factors are able to impact fat storage in ants, such as: latitude, environmental stressors, seasonal changes and carbohydrate intake (Elmes et al., 2013; Griffiths, 1991; Dussutour et al., 2016; Yang, 2006; Tschinkel, 1998). Additionally, it has been documented that ants with higher fat stores are able to survive adverse conditions better than those with lower fat stores (Dussutour et al., 2016). Meaning that increased fat storage in ants is likely a stress response to help promote survival. The ants in our experimental treatment were undergoing constantly increasing temperatures that caused great stress upon the colonies which is likely what led to the increase in fat storage observed in the larvae. However, many insects, including ants are known to increase fat, carbohydrate and protein stores in preparation for winter (Sinclair and Marshall, 2018; Kipyatkov, 2001; Lopatina, 2018; Ingram et al., 2009). During the time of experimentation, it was late autumn within the Northeastern USA meaning that the shift in seasons could have also caused the increase in fat storage within the ants due to their circadian clocks. Information regarding the fat storage within the larvae from our experiment was based on observations and no compositional data was collected. The analysis of fat storage increase was outside the scope of our experiment and future studies should investigate possible correlations between fat storage and heat stress in ants.

### Impact of warming temperatures on *A. rudis* bacterial communities

Few studies have looked at the impact that warming temperatures will have on *Aphaenogaster* ants, however none of them have documented the impact that these warming temperatures will have on their symbiotic bacterial communities. *Aphaenogaster* ants'

bacterial communities have just recently begun to be documented and studied (Pagalilauan et al., 2025; Kelleher and Ramalho, 2025). Which leaves many questions to be answered about what can impact their bacterial communities. The results of our study found that there is a significant difference in the composition of the bacterial communities of *A. rudis* based on temperature. Previous studies have documented the impact of warming temperatures on the bacterial communities of termites, beet armyworms and other ants (McMunn et al., 2022; Arango et al., 2021; Chen et al., 2022; Sapkota et al., 2024; Fan and Wernegreen, 2013). For example, McMunn et al. (2022) found that in *Cephalotes rohweri* the bacterial community composition varied with seasonal changes in temperatures and varied with laboratory temperature changes (McMunn et al., 2022). However, as mentioned previously, it was late autumn in the Northeastern USA while this experiment was occurring. Studies have investigated the influence of season on the bacterial communities of *Cephalotes* ants and termites (McMunn et al., 2022; Sapkota et al., 2024). These studies did find that the bacterial communities of ants and termites do change with the seasons, however these shift were attributed to changes in temperature associated with the seasons rather than the time of year itself (McMunn et al., 2022; Sapkota et al., 2024). This indicates that the changes we are observing in the bacterial communities of *A. rudis* is likely due to the experimental changes in temperature rather than the time of year we conducted our study.

Additionally, our study documented that there was an increase (non-significant) in the diversity of the bacterial community as temperatures increased. Previous studies have documented a decrease in the bacterial diversities of *Cephalotes rohweri* and *Camponotus* spp. in warmer temperatures compared to cooler ones (McMunn et al., 2022; Fan and Wernegreen, 2013), however we found other patterns in *Aphaenogaster*. Our results could be explained by the presence of *Wolbachia*. We observed a decrease in the presence of *Wolbachia* spp. in the experimental group compared to the control. This decrease in *Wolbachia* spp. in the experimental group could have left *A. rudis* more vulnerable to other bacteria compared to the control.

*Wolbachia* spp. have been previously documented in *Aphaenogaster* ants and a variety of other organisms such as: bees, wasps, mosquitos, bedbugs, termites, and other ants (Kelleher and Ramalho, 2025; Ramalho and Moreau, 2020; Ramalho et al., 2021; Kyei-Poku et al., 2006; Jiggins, 2017; Chebbah et al., 2023; Gerth et al., 2013; Duplouy et al., 2015; Lo and Evans, 2007). However, the function of *Wolbachia* spp. is open to debate. In *Monomorium pharaonis* infected with *Wolbachia* spp. they have faster population growth, produce more reproductive progeny, have an accelerated lifecycle with higher productivity, faster growth rates, higher survival, higher metabolic rates, and overall longer

lifespans (Singh et al., 2024; Singh and Linksvayer, 2020; Pontieri et al., 2017). *Wolbachia* spp. have also been documented to help with vitamin B synthesis in the ant *Tapinoma melanocephalum*, *Rhodnius prolixus*, and *Cimex lectularius* (Cheng et al., 2019; Filée et al., 2022 preprint; Hickin et al., 2022). In *Tapinoma melanocephalum*, *Wolbachia* spp. have been shown to produce B2 and B3 vitamins and synthesize nutrients needed by the host (Cheng et al., 2019). In *Rhodnius prolixus* (kissing bugs) and *Cimex lectularius* (bedbugs), *Wolbachia* spp. can regulate the production of B vitamins through the biotin operon (Filée et al., 2022 preprint; Hickin et al., 2022). The production of B vitamins by *Wolbachia* spp. for these organisms is because they are lacking these vitamins in their regular diets. This means that these organisms rely upon the synthesis of these B vitamins by *Wolbachia* (Cheng et al., 2019). Additionally, *Wolbachia* has also been shown to cause reproduction alterations, feminization, parthenogenesis, sperm-egg incompatibility, male sterility and a more female dominated population in *Ostrinira* sp., wasps, mosquitoes and mites (Werren et al., 2008; Saridaki and Bourtzis, 2010; Kageyama et al., 2002; Stouthamer et al., 1990; Weeks and Breeuwer, 2001; Dedeine et al., 2005; Moretti et al., 2018). This indicates that the presence of *Wolbachia* can have negative impacts on reproduction within insect populations. However, the function of *Wolbachia* spp. in *Aphaenogaster* ants is still ultimately unknown, but it could both be impacting their development and reproduction and be synthesizing essential B vitamins. This could indicate that *Aphaenogaster* rely upon their *Wolbachia* spp. symbionts.

We observed a decrease in the presence of *Wolbachia* spp. as temperatures began to rise. *Wolbachia* spp. are heat-sensitive bacteria and studies have shown changes in the presence of *Wolbachia* spp. in warm temperatures in mosquitoes, flies, moths, wasps, termites, mites and ants (Van Opijnen and Breeuwer, 1999; Sugimoto et al., 2015; Roy et al., 2015; Jeyaprakash and Hoy, 2010; Bouwma and Shoemaker, 2011; Nikoh et al., 2014; Ulrich et al., 2016). When the mosquito *Aedes aegypt* was subjected to warm temperatures, they experienced a decrease in the presence of *Wolbachia* spp. and a decrease in the maternal transmission of *Wolbachia* spp. (Mancini et al., 2021; Hague et al., 2020). In *Drosophila* it has been shown that *Wolbachia* spp. is able to impact the thermal tolerance of their hosts, forcing them to prefer cooler temperatures than uninfected individuals (Hurst et al., 2000; Clancy and Hoffmann, 1998; Bordenstein and Bordenstein, 2011). In the wasps, *Nasonia vitripennis* and *Leptopilina heterotoma*, and ants, such as *Anoplolepis gracilipes*, the presence of *Wolbachia* spp. decreases when temperatures begin to increase (Bordenstein and Bordenstein, 2011; Lin et al., 2023; Mouton et al., 2007). The drastic impact that temperature has on the presence of *Wolbachia* spp. within the host is due to the destruction of the bacterial cell membrane at warm temperatures and the decrease in the vertical transfer of *Wolbachia* spp. at warm temperatures (Ulrich et al., 2016; Mancini et al., 2021; Hague et al., 2020). Within *A. rudis* we observed a decrease in *Wolbachia* spp. as temperature began to rise and we also observed an increase in mortality and a decrease in brood presence. Our findings suggest that as temperatures begin to rise *A. rudis* will lose their *Wolbachia* spp. symbionts, which occurred simultaneously with increased mortality within the colony and a decrease in reproduction and development of brood. However, future studies should focus on the whole genome sequencing of these *Wolbachia* spp. to determine the real function of these bacteria within ants.

We also observed an increase in the presence of *Corynebacterium* sp. in warmer temperatures. *Corynebacterium* spp. have been observed in ants, such as *Trachymyrmex septentrionalis*, *Atta texana*, *Linepithema humile*, and *Leptogenys chinensis*, and studies

have shown that *Corynebacterium* spp. presence in ants is not common and is usually acquired due to unique circumstances (Ishak et al., 2011; Rajagopal et al., 2023; Meirelles et al., 2016; Lester et al., 2017). The function of *Corynebacterium* spp. in ants remains unstudied, however in humans *Corynebacterium* spp. can produce amino acids, antibiotics and cause infections (Bernard and Funke, 2015; Olender, 2012). The function of *Corynebacterium* spp. in ants could be similar, leading to infections and disease. However, transcriptomics on *Corynebacterium* sp. could help reveal the function it is playing in *Aphaenogaster*.

### The impact of development stage on *A. rudis* bacterial communities

Our study also investigated the differences in the bacterial communities of *A. rudis* brood (eggs, larvae, pupa) and workers (recently emerged workers and workers). We found that the bacterial community composition and abundance of the brood and workers within the control were not significantly different, however the bacterial community composition of the brood and workers for the experimental group were significantly different. The differences in the bacterial communities of ant development stages have been previously documented in *Aphaenogaster* ants and other ants such as *Decaton*, *Colobopsis*, and *Camponotus* (Ramalho et al., 2017b, 2020; Kelleher and Ramalho, 2025; Koto et al., 2020). We observed that *Wolbachia* spp. were driving this difference. *Wolbachia* has been shown to help with the growth and development of *Monomorium pharaonis*, with more *Wolbachia* being present in the brood (Singh et al., 2024; Singh and Linksvayer, 2020). We also observed a high abundance of *Entomoplasma* spp. within the workers than the brood across both the control and experimental treatments. *Entomoplasma* spp. have been identified in a variety of ants, such as *Trachymyrmex* spp., *Atta texana*, *Megalomyrmex* spp., *Eciton burchellii, Eciton vagans, Dorylus molestus*, and *Acromyrmex* spp. (Sapountzis et al., 2015; Allert, 2017; Meirelles et al., 2016; Liberti et al., 2015; Zani et al., 2021; Funaro et al., 2011; Zheng et al., 2022) and within *Megalomyrmex* and *Eciton* there has been shown to be a higher abundance of *Entomoplasma* within workers than brood (Zani et al., 2021; Funaro et al., 2011; Zheng et al., 2022). Additionally, *Entomoplasma* is commonly observed in predatory and generalist ants. Previous studies have speculated that *Entomoplasma* could help with the processing of chitin, a compound found within fungus and the cuticle of insects (Liberti et al., 2015). *Aphaenogaster* ants are generalists and *Entomoplasma* has been previously documented within their bacterial communities (Kelleher and Ramalho, 2025), so *Entomoplasma* could help with digestion within the workers. However further investigation into the function of these *Entomoplasma* spp. is required to make any definitive conclusions.

### Conclusion

Global climate change will cause drastic changes in temperature and weather events that will impact all living organisms, but how these changes influence host–bacterial interactions are important because of their contribution to global biodiversity. These interactions are essential to the survival of various organisms, such as ants. By better understanding how these interactions change and evolve in relation to changing temperatures we can predict how our biodiversity will be influenced in the future. Our study sought to understand how climate change influenced *Aphaenogaster* ants and their bacterial communities. *Aphaenogaster* ants are a globally distributed genus with many species native to North America. North American species are forest dwellers that are keystone seed distributors of woodland and grassland plants (Warren and Chick,

2013). Predicting the impact that global climate change will have on these ants is essential to determining their survival likelihood and implementing conservation efforts.

Our results show that mortality rates of *A. rudis* drastically increase at warm temperatures, and the presence of brood also decreases. Additionally, we found that warmer temperatures impact the composition of the symbiotic bacterial communities of *A. rudis* causing a decrease in the presence of *Wolbachia* spp. in warmer temperatures. The endosymbiont *Wolbachia* spp. have been known to assist with the growth and development of colonies as well as the production of the B vitamin nutrients. The loss of *Wolbachia* spp. due to the warm temperature in *A. rudis* could be impacting the survival and growth of the *A. rudis* colonies in this experiment. However, further investigation into the function of these *Wolbachia* species in *A. rudis* is needed to determine the impact that the change in the *A. rudis* bacterial communities will have on these ants. Our data suggests that as temperatures begin to climb *Aphaenogaster* ants will be negatively impacted as their mortality rates increase, and bacterial communities shift in the changing climate.

Several studies have been demonstrating how to mitigate the impact of climate change on insect populations through habitat rehabilitation/preservation and the reduction of pollution (Halsch et al., 2021; Kawahara et al., 2021; Smith et al., 2015; Sharma et al., 2023). These efforts consist of the maintenance and addition of insect friendly features, such as leaf litter (de Queiroz et al., 2013), log piles (Torgersen and Buff, 1995), and pollinator hotels (MacIvor and Packer, 2015), helps to protect insects from predation, provide homes and help regulate essential microclimates. Additionally, the use of pesticide and herbicides harms insect populations due to many of them being non-specific leading them to target many insect species (Kawahara et al., 2021; Samways et al., 2020). Reduction of the use of these chemicals will help with the recovery and survival of insect populations in the future (Kawahara et al., 2021; Samways et al., 2020). Overall, the results of this study demonstrate that *A. rudis* is extremely sensitive to heat. As temperatures continue to rise, their bacterial symbiosis is likely to shift, and mortality rates will increase. Therefore, it is crucial to implement conservation efforts to protect these important ants and other insects from the impacts of climate change.

## MATERIALS AND METHODS
### Ant collection and identification
For this study four queen right colonies for *A. rudis* were collected within decaying wood and soil using an insect aspirator from the Gordon Natural Area (temperate mix-hardwood forest) located at 39.9352°N, 75.5993°W in West Chester, PA, USA. The Institutional Animal Care and Use Committee (IACUC) does not require an animal use protocol for activities involving lower-level invertebrate species (e.g. Brine shrimp, fruit flies, ants, etc.). These colonies were then raised in the laboratory on a consistent diet of sugar water, apples, tuna and crickets at a controlled temperature of 22°C over the course of 6-12 months. Twenty-four *Aphaenogaster rudis* samples were collected from the four colonies raised at 22°C. This consisted of five eggs masses, five larvae, four pupa, five recently emerged workers and five workers. This made up the control group for the experiment. After collection samples were stored in 95% ethanol at −20°C until a DNA extraction was performed.

After samples were collected at 22°C, the same colonies were placed inside an incubator where the temperature was slowly raised from 22°C to 32°C over the course of nine weeks to avoid killing the colonies from a drastic change in temperature. The same colonies were used in the control and experimental treatments because previous studies have shown that the bacterial communities of ants, *Temnothorax* sp., *Trachymyrmex* sp., *Colobopsis* spp. and *Camponotus* spp., are influenced by colony membership (Green and Klassen, 2022; Segers et al., 2019; Ramalho et al., 2017b). Therefore, we

used the same ant colonies as the control and experimental treatments to account for this possible influence. The temperature was then maintained at 32°C for 5 weeks. The maximum temperature of 32°C was selected for this study because the average annual temperature in Pennsylvania is 21-25°C. The 2024 Pennsylvania Climate Impacts Assessment predicts that the average annual temperature will increase by 3.7°C by 2050, however it was also predicted that by 2050 Pennsylvania will begin to experience an increased number of days with temperatures reaching above 32°C (Pennsylvania Department of Environmental Protection, 2024). While ants were subjected to warming temperatures the mortality rates of adult workers were documented at each temperature. Growth and activity levels within the colonies were also monitored. After 5 weeks at our optimal experimental temperature of 32°C ants were collected. Twenty-five *Aphaenogaster* samples were collected from the four colonies. This consisted of five egg masses, five larvae, five recently emerged workers and ten workers. This made up the experimental group for the experiment. We were unable to collect pupa from the experimental group due to no pupa being present. After collection samples were stored in 95% ethanol at −20°C until a DNA extraction was performed.

To confirm the identity of the ants as *A. rudis* additional workers were collected from each colony, pinned and identified using morphology keys (Fisher and Cover, 2007; Mackay and Mackay, 2017). Samples were submitted to the entomology collection at West Chester University. To verify the identity of the ants COI barcoding was performed. DNA extraction and PCR was performed to amplify the CO1 gene (LCO1490 5′-GGTCAACAAATCATAAAGATATTGG-3′, and HCO2198 5′-TAAACTTCAGGGTGACCAAAAAATCA-3′) (Folmer et al., 1994). Samples were run using the following thermocycling program: initial denature at 94°C for 1 min, then 35 cycles of 94°C for 30 s, 48°C for 1 min, and 72°C for 2 min and the final extension at 72°C for 5 min. This was verified with gel electrophoresis. The PCR product was cleaned using 1.5 µL ExoSAP-IT per 25 µl of sample and was run in the thermocycle at 37°C for 15 min and then 80°C for 15 min. Samples were then sequenced at Azenta Life Sciences (South Plainfield, NJ, USA) and species was determined by using NCBI BLAST on the sequences (Table S1). Within NCBI BLAST, the nucleotide BLAST (BLASTn) setting was used on the default settings that looked at standard databases at the core nucleotide database (core_nt) and was optimized for highly similar sequences (megablast). Sequences were submitted to NCBI as a BioProject under the accession number PRJNA1247167.

Behavior of the ants within the colonies were observed daily over the course of the experiment to look for any changes. We observed the behaviors of foraging and speed of foraging within workers, maintenance of a clean colony (consistently using their discard piles), and the location and presence of brood within the colony. To determine whether there were behavioral changes, observations were made at both the control temperature (22°C) and experimental temperature (23-32°C) and compared. Additionally, one other colony of *Aphaenogaster* was in the laboratory being raised at 22°C, so comparisons could be made between the experimental colonies and those not undergoing any treatment. This allowed for us to make comparisons within a colony and across multiple colonies.

### DNA extraction and sequencing
Whole ants and whole brood were cleaned with DI water and 100% ethanol to remove external contaminating microbes. Ants were then shaken in a bead tube with Proteinase K overnight to efficiently break cell membranes (Moreau et al., 2014). DNA was then extracted from the whole ant using the DNeasy Blood & Tissue Kit (Qiagen) following manufacturer's instructions. Six blank samples were also included in the DNA extraction as negative controls.

Polymerase Chain Reactions (PCRs) and sequencing were performed at CD Genomics (Shirley, NY, USA). Two PCR reactions were run on each sample to amplify the bacterial 16S rRNA genes within the V3 and V4 regions. First a PCR to amplify the V3 and V4 regions of the 16 s rRNA gene was performed with adaptors on the primers 341 forward (F) 5′-ACACTCTTTCCCTACACGACGCTCTTCCGATCTCCTACGGGNGG-CWGCAG-3′, and 805 reverse (R) 5′-GTGACTGGAGTTCAGACG-TGTGCTCTTCCGATCTGACTACHVGGGTATCTAATCC-3′. Samples were then run using the following thermocycling program: initial denature at

98°C for 45 s, then 27 cycles of 98°C for 10 s, 60°C for 1 min, and 72°C for 30 s and the final extension at 72°C for 5 min. PCR product was verified using gel electrophoresis on a 2% agarose gel. Sequence preparation was performed by running a PCR using a i5 index PCR primer (5′-AATGATACGGCGACCACCGAGATCTACACXXXXXXXXXACACTCT-TTCCCTACACGACGCTCTTCCGATCT-3′) and a i7 index PCR primer (5′-CAAGCAGAAGACGGCATACGAGATXXXXXXXXXGTGACTGGA-GTTCAGACGTGTGCTCTTCCGATCT-3′). Samples were then run using the following thermocycling program: initial denature at 98°C for 1 min, then nine cycles of 98°C for 10 s, 60°C for 30 s, and 72°C for 30 s and the final extension at 72°C for 1 min. Quantitative library concentration was then determined using a Qubit dsDNA HS Assay kit. Amplicons were sequenced with an Illumina MiSeq using the 300-bp paired-end kit (version 3).

## Bioinformatic analysis

A demultiplex paired end analysis was performed in QIIME2 (version 3023.5) (Bolyen et al., 2019) after raw sequences were uploaded. Sequences were then trimmed (forward trim 5 and trunc 245, reverse trim 5 and trunc 240) and a feature table was created. DADA2 was then used to perform taxonomic classification using the SILVA 138 classier that identified bacteria into ASVs at 99% similarity (Callahan et al., 2016; Quast et al., 2012). The Decontam package in Qiime2 was then used to filter out bacterial contamination from DNA extraction or PCR from the samples using the negative controls. Decontam was also used to remove mitochondria and chloroplast. The align-to-tree-mafft-fasttree function in Qiime2 was used to create a phylogenetic tree for the ASVs using the fasttree functions internal rooting mechanism. This phylogenetic tree is required to run further bacterial community analysis in Qiime2.

PERMANOVA using the unifrac distance matrix were performed in Qiime2. Additionally, the post hoc pairwise function with multiple comparisons corrected by Bonferroni correction was performed. PERMANOVAs are widely accepted and used to analyze ASV data because this test analysis allows for non-parametric and compositional data (McMunn et al., 2022; Ramalho et al., 2020; Donhauser et al., 2020; Uribe-Lorío et al., 2019; Pagalilauan et al., 2025; Kelleher and Ramalho, 2025; Green and Klassen, 2022; Segers et al., 2019). A similarity percentage (SIMPER) analysis using the Bray-Curtis dissimilarity matrix was performed to verify ASVs contribution to the bacterial communities using the program Past4 (Hammer et al., 2001). SIMPER analyses were run because they determine and rank which taxa are driving the significant differences observed in community composition and abundance.

Alpha diversity measures of Shannon diversity (H) and Simpson's diversity index/Simpsons D were performed in Qiime2. Shannon diversity was determined using the faith_pd_vector and Simpson's diversity index/Simpsons D was determined using the Simpsons metric. The alpha diversity measure of Margalef's index was manually calculated from the ASV table. Shannon diversity was used to determine the species richness and evenness of the bacterial community, Simpon's Diversity/Simpsons D was used to determine if there were any dominant bacteria within the community, and Margalef's index was used to measure species richness on its own.

The logistic regression model and visualizations of network analysis, PCoA plots, nMDS diagrams and heat maps were performed in R (version 4.4.0) (R Core Team, 2024) using the package qiime2R (Bisanz, 2018), ggplot2 (Wickham, 2016), vegan (Oksanen et al., 2025), metacoder (Foster et al., 2017), tidyverse (Wickham et al., 2019), and phyloseq (McMurdie and Holmes, 2013). The logistic regression model was used on the non-linear mortality data to determine whether the mortality rates differed among the colonies over time. The logistic regression model was fit using the glm() function with family=binomial and the emmeans package was used to perform pairwise comparisons (Lenth, 2024). After decontamination we obtained 7951 ASVs and 1,393,873 reads. Analysis resulted in the loss of two samples from the control treatment due to low number of reads. Raw sequences are available on NCBI SRA with the access number PRJNA1223423.

## Statements and declarations

We declare that we did not use generative artificial intelligence (AI) tools in the composition of our manuscript, or did any part of this work involve the use of such tools.

## Acknowledgements
We would like to acknowledge the Ramalho lab and Dr Rachelle Adams for their support in completing this work. We would like to thank the Biology department at West Chester University for their support and use of equipment. We would also like to thank Amanda Cohan, Kay Mcfadden, Alexandra Gianaris, Isabella Barboza de Almeida, and Ethan Scolastico for assisting with *Aphaenogaster* care and laboratory work. L.K. would like to thank Timothy Hilferty, Michelle Donnelly-Kelleher and Christopher Kelleher for their unconditional support.

## Competing interests
The authors declare no competing or financial interests.

## Author contributions
Conceptualization: L.K., M.O.R.; Data curation: L.K.; Formal analysis: L.K.; Funding acquisition: M.O.R.; Investigation: L.K.; Methodology: L.K., M.O.R.; Project administration: L.K.; Resources: M.O.R.; Supervision: M.O.R.; Visualization: L.K.; Writing – original draft: L.K.; Writing – review & editing: L.K., M.O.R.

## Funding
This work was supported by PRG (7513312112), TRIANO award, West Chester University. Open Access funding provided by The Ohio State University. Deposited in PMC for immediate release.

## Data and resource availability
The data that support the findings of this study are openly available in NCBI at https://www.ncbi.nlm.nih.gov/bioproject/PRJNA1223423/, reference number PRJNA1223423 and at https://www.ncbi.nlm.nih.gov/bioproject/PRJNA1247167/, reference number PRJNA1247167.

## Ethics approval
The IACUC (Institutional Animal Care and Use Committee) does not require an animal use protocol for activities involving lower-level invertebrate species (e.g. Brine shrimp, fruit flies, ants, etc.)

## Peer review history
The peer review history is available online at https://journals.biologists.com/bio/lookup/doi/10.1242/bio.062145.reviewer-comments.pdf

## First Person
This article has an associated First Person interview with the first author of the paper.

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
