## [Peer Review File · Biology Open]

Impact of Rising Temperatures on the Bacterial Communities of Aphaenogaster Ants

Lily Kelleher and Manuela Ramalho
DOI: 10.1242/bio.062145

Editor: Kendra J. Greenlee

Review timeline

Original submission:	13 June 2025
Editorial decision:	19 June 2025
Resubmission:	1 July 2025
Editorial decision:	2 July 2025
Resubmission:	4 July 2025
Editorial decision:	9 July 2025
First revision:	9 July 2025
Accept:	18 July 2025

Original submission

First decision letter

MS ID#: bio.062113

MS TITLE: Impact of Rising Temperatures on the Bacterial Communities of Aphaenogaster Ants

AUTHORS: Lily Kelleher; Manuela Ramalho

I am writing to let you know that I have now reached a decision on the above manuscript. I am afraid that, after careful consideration, I feel that it cannot currently be accepted for publication in Biology Open.

The reviewer reports are shown at the bottom of this email or can be accessed, together with a copy of this decision letter, by going to:

As you will see, the reviewers raise a number of substantial criticisms that prevent me from accepting your paper for publication. I do hope you find the reviewer comments helpful in allowing you to revise the manuscript for successful submission elsewhere.

Reviewer 1

Comments for the author

This study investigates how warming influences Aphaenogaster ant survival and the ants' bacterial communities. The paper is generally well written and the analyses are mostly well executed (except where noted below), but unfortunately, the study has a fatal flaw in terms of experimental design. The authors studied four colonies of ants maintained first at 22 degrees C and then all four colonies were warmed to 32 degrees C. Thus, the study design lacks appropriate controls (i.e., colonies kept at 22 degrees C throughout the experiment). It appears from Lines 188-191 (and later in the analysis of worker survival, e.g., in Figure 1) that a few other colonies were indeed kept at

22 degrees C throughout the study and therefore could serve as true controls. However, my read of the 16S rRNA amplicon portion of the study is that these true controls did not have their bacterial communities sequenced. Instead, the "experimental" and "control" colonies in the 16S portion of the study were the same colonies at different time points (and temperatures), such that the "control" colonies are not true controls. I don't see a way around this, unfortunately, and so I ticked "no" for "Does each figure have the appropriate controls?" This same problem also led me to tick "no" for "Are the manuscript's conclusions supported by the data?" and for "Do the authors cite and discuss the merits of data that would argue for and against their conclusion?" because the authors do not discuss this problem anywhere. The problem with this design is that it is impossible to say if warming per se produced the observed effects or if the effects are because of, for example, spending a greater length of time under laboratory conditions (e.g., a longer time on the lab diet), or greater ant colony age, or a different time of year, etc. I also ticked "no" for "Were experiments performed using adequate number of biological replicates?" because even when true controls were included, all the ants came from just one control colony (e.g., the data in Figure 1).

Given my concerns about the experimental design, I also thought the authors' claims in the title and abstract were unsupported, leading me to tick "no" for "Does the manuscript title & abstract accurately reflect the contents of the manuscript, without hyperbole?"

I also ticked "no" for "Were the data analyzed using appropriate statistical tests," but I regard this as a fixable problem. I had a few recommendations for best ways to analyze the data, but what the authors did is mostly correct, with just a few, relatively minor, exceptions. Specifically, I think it makes more sense to use a linear model than several t-tests to test whether colony identity predicts ant mortality rate (e.g., in Lines 257-263 and Figure 1), and then posthoc comparisons to determine whether your colony raised at 22 degrees is different from your other 4 colonies. Comparing each experimental colony to the control colony with a t-test does not maintain an appropriate experiment-wide Type 1 error rate. This is easily fixed in a revision. Also, I could not figure out the PERMANOVA results reported in Lines 358-363. Why is there a separate p-value for control and experimental colonies, rather than a single p-value testing for the treatment effect of control vs. experimental colonies? PERMANOVA is also tricky here because colony ID should really be included as a random effect, and I believe it is not possible to include random effects in PERMANOVA models fit in R (to my knowledge, anyways). Still, colony ID should be included as a fixed effect.

Regarding the discussion of Wolbachia in Lines 491-507, I was surprised that the authors did not mention that Wolbachia are classically considered parasites of their insect hosts, because they can lead to male sterility, etc. I appreciate that there is evidence that they can be beneficial in ants and even other insects, as the authors describe in the Discussion, but I felt this perspective was a bit one-sided.

Minor editorial comments:

Line 69: The influence of temperature on what exactly?

Line 83: "ants' ability" is missing an apostrophe

Line 85: the "and" in the sentence is unnecessary

Line 140-142: Did you collect the queen for each colony, and did she survive until the end of the study?

Line 156: "predicts" is missing an s

Line 193: should read "each larva" (no e on larva, singular)

Line 235: I think the PERMANOVA should have used either Bray-Curtis or UniFrac but not both

Line 289: "comprised" not "compiled" and ditto for Line 294

Line 431: Would be good to add some citations in support of this sentence

Line 493: "open to" not "up to" debate, I think

Line 534: "could" should be lowercase

Reviewer 2

Comments for the author

In this manuscript, Kelleher and Ramalho investigated the impact of elevated temperatures on the bacterial communities of *Aphaenogaster rudis* ants. For this purpose, the authors used 4 colonies collected around West Chester area and assessed the survival of ants at control (22 °C) and experimental (32 °C) temperatures. Bacterial community profiling via 16s rRNA amplicon sequencing was performed on different developmental stages under two temperature treatments to investigate the influence of temperature on bacterial communities. Ants exhibited increased mortality rates and reduced brood production at 32 °C as compared to control temperature. Elevated temperature also affected bacterial communities of ants leading to the decrease of *Wolbachia* spp. and an increase of *Corynebacterium* sp. This study provides an example of the negative impact of global climate change on insect species and their symbionts. While overall the conclusions of this work are supported by the results, clarifications of methodological details are needed to fully appreciate the data.

- It should be explicitly specified what a sample is. Is it a single ant, larvae, etc? I assume it is. However, I am confused with the numbers specified in methods and shown in figures. For example, in methods, it is written that 24 samples were collected for control, but figures (Figure 2 for example) show only 22 columns. Such discrepancies should be explained (e.g. where are the missing samples?). Also, showing pictures of ants is not really helpful in assigning a specific column to a sample. The author should do it in a better way.
- Fat storage quantification (L192-197 and Figure S2). First, is this microscopy-base technique an acceptable method in the field to measure fat storage? Can the authors provide references? Second, quantitative data should be provided to support the claim of increased fat storage. Third, from the images in Fig. S1, it is not clear what fat stores are. The authors could add arrows to point out these fat stores on images. More details must be added to the Figure S1 legend, explaining to the readers what this figure is about and where fat stores are.
- Methods lack description of sequencing library preparation and sequencing details
- It seems that with amplicon sequencing the authors managed to differentiate several *Wolbachia* sp. Could the authors specify the parameters they used to assign *Wolbachia* to a different species?
- L467-469 - how can the authors exclude that such shift in the seasons was not affecting microbiota? Maybe it is the season and not the temperature? This is an important point that requires at least discussion
- Manuscript requires proofreading. Here are just few examples: L107- "are" is missing after "in", L401 - "between" should be deleted, L530- "super" is not a scientific language, title of table S1- "of" after *rudis* is not needed
- Reference list should be formatted: see, for example 106, 115
- L69 - influence on what?

Reviewer's Responses to Questions

Experimental quality

Does each figure have the proper controls?

If 'No', please indicate reasons in Comments for Author box below.

Reviewer #1:

- No

Reviewer #2:

- Yes

Were the data analyzed using appropriate statistical tests?

If 'No', please indicate reasons in Comments for Author box below.

Reviewer #1:

- No

Reviewer #2:

- Yes

Reproducibility

Were experiments performed using adequate number of biological replicates?

If 'No', please indicate reasons in Comments for Author box below.

Reviewer #1:

- No

Reviewer #2:

- Yes

Does the methods section provide sufficient detail to permit reproducibility?

If 'No', please indicate reasons in Comments for Author box below.

Reviewer #1:

- Yes

Reviewer #2:

- No

Completeness

Are the manuscript's conclusions supported by the data?

If 'No', please indicate reasons in Comments for Author box below.

Reviewer #1:

- No

Reviewer #2:

- Yes

Scholarship

Do the authors cite and discuss the merits of data that would argue for and against their conclusion?

If 'No', please indicate reasons in Comments for Author box below.

Reviewer #1:

- No

Reviewer #2:

- Yes

Does the manuscript title & abstract accurately reflect the contents of the manuscript, without hyperbole?

If 'No', please indicate reasons in Comments for Author box below.

Reviewer #1:

- No

Reviewer #2:

- Yes

Resubmission

MS ID#: bio.062136

MS TITLE: Impact of Rising Temperatures on the Bacterial Communities of Aphaenogaster Ants

AUTHORS: Lily Kelleher; Manuela Ramalho

Author response to reviewers' comments

Reviewer 1:

This study investigates how warming influences Aphaenogaster ant survival and the ants' bacterial communities. The paper is generally well written and the analyses are mostly well executed (except where noted below), but unfortunately, the study has a fatal flaw in terms of experimental design. The authors studied four colonies of ants maintained first at 22 degrees C and then all four colonies were warmed to 32 degrees C. Thus, the study design lacks appropriate controls (i.e., colonies kept at 22 degrees C throughout the experiment). It appears from Lines 188-191 (and later in the analysis of worker survival, e.g., in Figure 1) that a few other colonies were indeed kept at 22 degrees C throughout the study and therefore could serve as true controls. However, my read of the 16S rRNA amplicon portion of the study is that these true controls did not have their bacterial communities sequenced. Instead, the "experimental" and "control" colonies in the 16S portion of the study were the same colonies at different time points (and temperatures), such that the "control" colonies are not true controls. I don't see a way around this, unfortunately, and so I ticked "no" for "Does each figure have the appropriate controls?" This same problem also led me to tick "no" for "Are the manuscript's conclusions supported by the data?" and for "Do the authors cite and discuss the merits of data that would argue for and against their conclusion?" because the authors do not discuss this problem anywhere. The problem with this design is that it is impossible to say if warming per se produced the observed effects or if the effects are because of, for example, spending a greater length of time under laboratory conditions (e.g., a longer time on the lab diet), or greater ant colony age, or a different time of year, etc. I also ticked "no" for "Were experiments performed using adequate number of biological replicates?" because even when true controls were included, all the ants came from just one control colony (e.g., the data in Figure 1). Given my concerns about the experimental design, I also thought the authors' claims in the title and abstract were unsupported, leading me to tick "no" for "Does the manuscript title & abstract accurately reflect the contents of the manuscript, without hyperbole?"

Response: Thank you for noticing this discrepancy within the methods. We chose to do this because previous studies on ant bacterial communities have revealed that colony membership can influence the bacterial communities, so we wanted to control for this possible influence, so we used the same colonies as our control and experimental treatments. To accept your suggestions we have added a portion to the methods to help make this more clear to the reader. Please refer to L153-L157. We have also added an acknowledgement for the usage of one "Control" colony for the mortality analysis. Please Refer to L266-L267. Thank you.

I also ticked "no" for "Were the data analyzed using appropriate statistical tests," but I regard this as a fixable problem. I had a few recommendations for best ways to analyze the data, but what the authors did is mostly correct, with just a few, relatively minor, exceptions. Specifically, I think it makes more sense to use a linear model than several t-tests to test whether colony identity predicts ant mortality rate (e.g., in Lines 257-263 and Figure 1), and then posthoc comparisons to determine whether your colony raised at 22 degrees is different

from your other 4 colonies. Comparing each experimental colony to the control colony with a t-test does not maintain an appropriate experiment-wide Type 1 error rate. This is easily fixed in a revision.

Response: Thank you for this suggestion. We have changed this and ran an ANOVA with a Tukey's HSD post hoc test and corrected this analysis. Please refer to L264-L270 and Table S2. Thank you.

Also, I could not figure out the PERMANOVA results reported in Lines 358-363. Why is there a separate p-value for control and experimental colonies, rather than a single p-value testing for the treatment effect of control vs. experimental colonies? PERMANOVA is also tricky here because colony ID should really be included as a random effect, and I believe it is not possible to include random effects in PERMANOVA models fit in R (to my knowledge, anyways). Still, colony ID should be included as a fixed effect.

Response: Thank you very much for your suggestion. We looked at the differences between colonies for both the control and experimental groups separately because we wanted to look at this variable without the influence of treatment being present. We have made this clarification within the text. Please refer to L368-L371. Thank you.

Regarding the discussion of *Wolbachia* in Lines 491-507, I was surprised that the authors did not mention that *Wolbachia* are classically considered parasites of their insect hosts, because they can lead to male sterility, etc. I appreciate that there is evidence that they can be beneficial in ants and even other insects, as the authors describe in the Discussion, but I felt this perspective was a bit one-sided.

Response: Thank you for this suggestion. We have added more information about the presence of *Wolbachia* causing a female dominated population and male sterility in insects. Please refer to L522-L525. Thank you.

Minor editorial comments:

Line 69: The influence of temperature on what exactly?

Response: Fixed. Thank you.

Line 83: "ants' ability" is missing an apostrophe

Response: Fixed. Thank you.

Line 85: the "and" in the sentence is unnecessary

Response: Removed. Thank you.

Line 140-142: Did you collect the queen for each colony, and did she survive until the end of the study?

Response: Clarified. Thank you.

Line 156: "predicts" is missing an s

Response: Added. Thank you.

Line 193: should read "each larva" (no e on larva, singular)

Response: Removed the e. Thank you.

Line 235: I think the PERMANOVA should have used either Bray-Curtis or UniFrac but not both

Response: Thank you for noticing this error. This has been corrected.

Line 289: "comprised" not "compiled" and ditto for Line 294

Response: Changed. Thank you.

Line 431: Would be good to add some citations in support of this sentence

Response: Citations added. Thank you.

Line 493: "open to" not "up to" debate, I think

Response: Changed. Thank you.

Line 534: "could" should be lowercase

Response: Fixed. Thank you.

Reviewer 2:

In this manuscript, Kelleher and Ramalho investigated the impact of elevated temperatures on the bacterial communities of *Aphaenogaster rudis* ants. For this purpose, the authors used 4 colonies collected around West Chester area and assessed the survival of ants at control (22 °C) and experimental (32 °C) temperatures. Bacterial community profiling via 16s rRNA amplicon sequencing was performed on different developmental stages under two temperature treatments to investigate the influence of temperature on bacterial communities. Ants exhibited increased mortality rates and reduced brood production at 32 °C as compared to control temperature. Elevated temperature also affected bacterial communities of ants leading to the decrease of *Wolbachia* spp. and an increase of *Corynebacterium* sp. This study provides an example of the negative impact of global climate change on insect species and their symbionts. While overall the conclusions of this work are supported by the results, clarifications of methodological details are needed to fully appreciate the data.

- It should be explicitly specified what a sample is. Is it a single ant, larvae, etc? I assume it is. However, I am confused with the numbers specified in methods and shown in figures. For example, in methods, it is written that 24 samples were collected for control, but figures (Figure 2 for example) show only 22 columns. Such discrepancies should be explained (e.g. where are the missing samples?). Also, showing pictures of ants is not really helpful in assigning a specific column to a sample. The author should do it in a better way.

Response: Thank you for this suggestion. Refer to L148-L149 and L166-L167 within the methods for descriptions on what each sample is. We have added a statement about the two samples that were lost due to low numbers of reads. Please refer to L252-L253.

- Fat storage quantification (L192-197 and Figure S2). First, is this microscopy-base technique an acceptable method in the field to measure fat storage? Can the authors provide references? Second, quantitative data should be provided to support the claim of increased fat storage. Third, from the images in Fig. S1, it is not clear what fat stores are. The authors could add arrows to point out these fat stores on images. More details must be added to the Figure S1 legend, explaining to the readers what this figure is about and where fat stores are.

Response: Thank you for this suggestion. Previous publications have used microscopy-based techniques and ImageJ to analyze fat storage and lipid storage. Please refer to the references listed below. Quantitative data in regards to differences in fat storage can be found in L286-L290. This reports the Average % fat and a wilcoxon rank sum test showing that the fat storage is different. More detail has been added to figure S1 and the legend.

1. Fouad AD, Pu SH, Teng S, Mark JR, Fu M, Zhang K, Huang J, Raizen DM, Fang-Yen C. Quantitative Assessment of Fat Levels in *Caenorhabditis elegans* Using Dark Field Microscopy. *G3 (Bethesda)*. 2017 Jun 7;7(6):1811-1818. doi: 10.1534/g3.117.040840. PMID: 28404661; PMCID: PMC5473760.

2. Klapper, M., Ehmke, M., Palgunow, D., Böhme, M., Matthäus, C., Bergner, G., ... & Döring, F. (2011). Fluorescence-based fixative and vital staining of lipid droplets in *Caenorhabditis elegans* reveal fat stores using microscopy and flow cytometry approaches. *Journal of lipid research*, 52(6), 1281-1293.
3. T. Hellerer, C. Axäng, C. Brackmann, P. Hillertz, M. Pilon, & A. Enejder, Monitoring of lipid storage in *Caenorhabditis elegans* using coherent anti-Stokes Raman scattering (CARS) microscopy, *Proc. Natl. Acad. Sci. U.S.A.* 104 (37) 14658-14663, <https://doi.org/10.1073/pnas.0703594104> (2007).
4. Ugrankar-Banerjee, R., Tran, S., Bowerman, J., Kovalenko, A., Paul, B., & Henne, W. M. (2023). The fat body cortical actin network regulates *Drosophila* inter-organ nutrient trafficking, signaling, and adipose cell size. *Elife*, 12, e81170.
5. Hellerer T, Axäng C, Brackmann C, Hillertz P, Pilon M, Enejder A. Monitoring of lipid storage in *Caenorhabditis elegans* using coherent anti-Stokes Raman scattering (CARS) microscopy. *Proc Natl Acad Sci U S A.* 2007 Sep 11;104(37):14658-63. doi: 10.1073/pnas.0703594104. Epub 2007 Sep 5. PMID: 17804796; PMCID: PMC1976189.
6. Yen K, Le TT, Bansal A, Narasimhan SD, Cheng JX, Tissenbaum HA. A comparative study of fat storage quantitation in nematode *Caenorhabditis elegans* using label and label-free methods. *PLoS One.* 2010 Sep 16;5(9):e12810. doi: 10.1371/journal.pone.0012810. PMID: 20862331; PMCID: PMC2940797.

- Methods lack description of sequencing library preparation and sequencing details

Response: Library prep was performed using 16s rRNA sequencing and the i7 and i5 index primers. Please refer to L219-L227 for more details. Details about the sequencer used have been added. Refer to L227-L228. Thank you.

- It seems that with amplicon sequencing the authors managed to differentiate several *Wolbachia* sp. Could the authors specify the parameters they used to assign *Wolbachia* to a different species?

Response: Taxonomic classification of sequences was performed using the SILVA 138 classifier that identified bacteria into ASVs at 99% similarity. The SILVA 138 classifier identified different *Wolbachia* sp. present within the data, however they can only be classified as *Wolbachia* sp. 1, sp. 2, etc. because this is all the data that the classifier contains. However, we could determine that we are looking at multiple *Wolbachia* species. Please refer to L234-L235 for details in taxonomic classification.

- L467-469 - how can the authors exclude that such shift in the seasons was not affecting microbiota? Maybe it is the season and not the temperature? This is an important point that requires at least discussion

Response: Thank you for pointing this out. We have added a statement regarding this possible influence. Please refer to L494-L501.

- Manuscript requires proofreading. Here are just few examples: L107- "are" is missing after "in", L401 - "between" should be deleted, L530- "super" is not a scientific language, title of table S1- "of" after rudis is not needed

Response: Thank you. These have been fixed.

- Reference list should be formatted: see, for example 106, 115

Response: Thank you. All references have been checked for formatting.

- L69 - influence on what?

Response: Thank you. This sentence has been changed. Please refer to L69.

ResubmissionFirst decision letter

MS ID#: bio.062136

MS TITLE: Impact of Rising Temperatures on the Bacterial Communities of Aphaenogaster Ants

AUTHORS: Lily Kelleher; Manuela Ramalho

I have now reached a decision on the above manuscript. I appreciate the edits that were made to improve the manuscript. However, there are a number of significant issues remaining that prevent me accepting it for publication.

- 1) The analysis of mortality is confusing. The figure presents percent mortality, which should be analyzed using logistic regression or some type of survival analysis, but the results text states that ANOVA was used.
- 2) None of the references cited in the response to reviewers for justifying the methods used to analyze fat storage in the ants used the same method that this study uses. All of the references used either some sort of staining method or special microscopy techniques (e.g., coherent anti-Stokes Raman scattering (CARS) microscopy).
- 3) The results, lines 271-284, discuss the amount and locations of brood within the colonies. However, there are no data presented and no statistical analyses that support these claims.
- 4) I recommend adding a statistical analysis section to the methods to describe and justify the tests that were run. This could help clarify the reviewer concerns about the PERMANOVA test.
- 5) Table S3 does not match what is in the results section regarding the fat storage on lines 288-290. How was that statistical test selected? Line 379 states that Table S3 contains the PERMANOVA results. Table S2 is missing.

Having said that, should you be able to address the remaining concerns, then I would be happy to see the paper again, as a new submission. If after considering the feedback, you instead decide to submit elsewhere, please let me know, so that we can close our file.

Resubmission

MS ID#: bio.062145

MS TITLE: Impact of Rising Temperatures on the Bacterial Communities of Aphaenogaster Ants

AUTHORS: Lily Kelleher; Manuela Ramalho

First decision letter

I appreciate the changes you made to the manuscript in response to my previous comments. However, I still am confused about the mortality analysis. I was going to look at the data to understand how you ran the statistical test, but I wasn't able to find your mortality data in the supplemental material or in the online repository. Although we do not require authors to publish all of the data, we do encourage it. See our policy here (<https://journals.biologists.com/bio/pages/journal-policies#data>). The paper that you cite for the linear regression used mortality as a factor not as the measurement. I assume that you calculated

the percent mortality using the population size and the number of ants that were found dead each day. The data shown in Figure 1 do not look linear, which is an assumption of a linear regression.

I also wanted to clarify my comment about the fat storage data. My intention was not to have you remove the data, but rather to justify it using some other published morphological study stating how you know that the structures you were measuring are the fat bodies. Lastly, if you prefer to keep the fat storage data out of the manuscript, then it should be moved from the results section to the discussion (lines 297-301), and a similar action should be taken for the results section that talks about brood relocations (lines 287-296).

Author response to Editor comments

Response: Thank you. We have moved all information regarding the fat storage and the behavior involving the brood to only the discussion section of the manuscript. Please refer to L441-450.

Thank you very much for your comments on the mortality data. This is our first time processing this type of data and we are still learning the proper way to do so. We have changed to a logistic regression model to process this data, and we have made this change to our manuscript. We appreciate your feedback and guidance on choosing the correct approach. Please refer to L269-276, L289-297 and Table S2.

Second decision letter

MS ID#: bio.062145

MS TITLE: Impact of Rising Temperatures on the Bacterial Communities of *Aphaenogaster* Ants

AUTHORS: Lily Kelleher; Manuela Ramalho

I am happy to tell you that your manuscript has been accepted for publication in Biology Open, pending our standard publication integrity checks. It was accepted on 18 July 2025. Thank you for the additional changes that you made to the manuscript.